# On the Quenching Mechanism of Ce, Tb Luminescence and Scintillation in Compositionally Disordered (Gd, Y, Yb)$_3$Al$_2$Ga$_3$O$_{12}$ Garnet Ceramics

**Valery Dubov** [1] , **Daria Kuznetsova** [1] , **Irina Kamenskikh** [2], **Ilia Komendo** [1] , **Georgii Malashkevich** [3],
**Andrei Ramanenka** [3] , **Vasili Retivov** [1] , **Yauheni Talochka** [4], **Andrei Vasil'ev** [5] and **Mikhail Korzhik** [1,4,*]

1    National Research Center "Kurchatov Institute", 123098 Moscow, Russia; valery_dubov@mail.ru (V.D.);
     daria_kyznecova@inbox.ru (D.K.); i.comendo@gmail.com (I.K.); vasilii_retivov@mail.ru (V.R.)
2    Faculty of Physics, Lomonosov Moscow State University, 119991 Moscow, Russia; ikamenskikh@bk.ru
3    B.I. Stepanov Institute of Physics, National Academy of Science of Belarus, Nezalezhnastsi Av.68-2,
     220072 Minsk, Belarus; g.malashkevich@ifanbel.bas-net.by (G.M.); a.ramanenka@ifanbel.bas-net.by (A.R.)
4    Institute for Nuclear Problems, Belarus State University, 11 Bobruiskaya, 220030 Minsk, Belarus;
     yauheni.talochka@gmail.com
5    Skobeltsyn Institute for Nuclear Physics, Lomonosov Moscow State University, 119991 Moscow, Russia;
     anvasiliev52@gmail.com
*    Correspondence: mikhail.korzhik@cern.ch

**Abstract:** Two series of (Gd, Y, Yb)$_3$Al$_2$Ga$_3$O$_{12}$ quintuple compounds with a garnet structure and solely doped with Ce and Tb were prepared in the form of ceramics by sintering in oxygen at 1600 °C for two hours and studied for the interaction of activator ions with ytterbium ions entering the matrix. It was shown that the photoluminescence and scintillation of Ce$^{3+}$ ions are completely suppressed, predominantly by tunneling ionization through the charge transfer state (CTS) of the Ce$^{4+}$-Yb$^{2+}$ ions. The photoluminescence of Tb$^{3+}$ ions is quenched in the presence of ytterbium, but not completely due to the poor resonance conditions of Tb$^{3+}$ intraconfiguration transitions and the CTS of the single Yb$^{3+}$ and the CTS of Ce$^{4+}$-Yb$^{2+}$ ions. The scintillation in the visible range of both Ce$^{3+}$- and Tb$^{3+}$-doped samples is intensely quenched as well, which indicates strong competition from Yb$^{3+}$ ions to activators in interaction with the Gd substrate.

**Keywords:** ceramics; garnet; cerium; terbium; luminescence; scintillation; luminescence quenching

## 1. Introduction

Ytterbium Yb ions have been utilized as promising activator ions for photonic devices for decades [1–4]. At the same time, in other branches that use activator ions to obtain luminescence, in particular scintillators, it has not yet been possible to use the advantages introduced by this cation into the crystalline matrix. First, we note the high atomic number of ytterbium, which occupies a position in front of lutetium in the periodic table of elements. It is suitable for the formation of crystalline matrices with a high stopping power for ionizing radiation. Next, ytterbium is one of the cheapest rare earth elements (RE), which implies a significant reduction in the cost of crystalline material based on it, for example, in comparison with lutetium or even gadolinium.

Ytterbium, in the trivalent state Yb$^{3+}$ (f$^{13}$), has optical transitions in the near-infrared (NIR) range: absorption due to the transition from the lower Stark components of the ground term $^2$F$_{7/2}$ to the excited state $^2$F$_{5/2}$ and emission due to the transition from the lower Stark component of the term $^2$F$_{5/2}$ to the levels of the term $^2$F$_{7/2}$ [5]. The decay time of NIR luminescence, due to the forbidden transition in parity and multiplicity, is in the millisecond range of 1–2 ms depending on the host matrices. The yield of scintillations in the NIR range of Yb$^{3+}$ ions is quite high and amounts to no less than 90,000 photons/MeV in crystals with a garnet structure [6]. In the compounds with a capability for isovalent

substitution by $Yb^{3+}$ ions, the charge transfer transition (CTT) luminescence is observed at low temperatures [7], which is due to the radiative relaxation of electrons from the CTS into the valence band. The CTT luminescence has fast kinetics in the range of tens of ns and can provide high scintillation parameters for materials of the garnet or perovskite structural type at cryogenic temperatures [8–13]. The properties of the CTT of $Yb^{3+}$ ions, as well as those of other RE ions, are considered in detail in [14].

The ytterbium ion is a heterovalent ion. With nonisovalent doping using ytterbium ions, they can be stabilized in the divalent $Yb^{2+}$ state by another heterovalent impurity during synthesis under reducing conditions, as demonstrated in [15]. In a case of isovalent substitution [16,17], the $Yb^{2+}$ ion exhibits interconfigurational luminescence $4f^{13}5d^1 \rightarrow f^{14}$, which ensures a high scintillation yield in alkali halide materials based on iodine I [18–20]. In other compounds, it may be completely absent [21]. In self-activated scintillation compounds based on divalent ytterbium, although scintillation is fast, it has a low scintillation yield, less than 3000 ph/MeV [22].

The intracenter transitions involving the $^2F_{7/2}$ and $^2F_{5/2}$ terms of $Yb^{3+}$ have an energy of about 1.2 eV, which indicates the complete absence of resonance with the radiative transitions of the widely used activator ions Ce and Pr to create scintillators. However, both the first experiments on the doping of ytterbium phosphates and oxyorthosilicate with Ce ions [23,24] and recent experiments on the localization of $Ce^{3+}$ and $Pr^{3+}$ ions in $Yb_3Al_5O_{12}$ [25–27] showed strong quenching of the luminescence of activators at room temperature. No typically intense luminescence bands of activator ions were observed, and insignificant scintillation properties were exclusively due to transitions involving matrix ions. It can be concluded that $Yb^{3+}$ ions are a very effective quencher of the luminescence of traditional activator ions used to create fast and bright scintillators. The effect of luminescence and scintillation quenching was also observed in self-activated $Bi_4Ge_3O_{12}$ scintillators upon activation with ytterbium ions [28].

The authors of [29] considered the mechanism of $Ce^{3+}$ luminescence quenching by using a co-doping additive of ytterbium ions in $YAlO_3$: Ce, Yb crystals. For the first time, an assumption was made about the tunneling mechanism of the quenching the intracenter $Ce^{3+}$ luminescence using ytterbium ions. Subsequently, a model for such quenching was developed [30]. The authors of [31] considered the quenching mechanism when the nonradiative transition of $Ce^{3+}$ ions is due to the cooperative mechanism: the emission of two NIR luminescence quanta of $Yb^{3+}$ ions.

Contrary to the $Ce^{3+}$ ion, trivalent terbium ion $Tb^{3+}$ provides slow scintillations. However, their yield significantly exceeds the yield of scintillations when the compound is doped with Ce [32]. Therefore, along with their use in detectors coupled to the photosensors operating in a current mode, they are promising for other applications, for example, in indirect converters of radiation from isotope sources [33–35]. The authors of [36] studied $Yb_xY_{1-x}PO_4$:Tb emission, excitation, and time-resolved luminescence. They showed the occurrence of energy transfer from the $^5D_4$ level of $Tb^{3+}$ to $Yb^{3+}$ ions, followed by the emission of two NIR photons. This finding indicates an effective transfer of the luminescence spectral density of emitted photons in compounds containing $Tb^{3+}$ and $Yb^{3+}$ ions to the NIR spectral region.

With the results gathered to date, one can note that the study was carried out in relatively simple crystalline systems. Apparently, the incorporation of the $Gd^{3+}$ ions into the matrix can introduce additional features into the crystalline system. An interaction of $Gd^{3+}$ and $Yb^{3+}$ ions in the crystalline compound may result even in three-photon NIR quantum cutting [37]. When excited with ionizing radiation, the migration of holes along the $Gd^{3+}$ sublattice and their delivery to ytterbium ions can prevent phosphorescence. Finally, Gd ions can compete with Yb ions to catch electrons and to resist complete quenching of luminescence and scintillation in the material in the visible range. In this context, combining terbium ions with a high excitation efficiency [38] and ytterbium ions in the same material promises obvious advantages both in terms of enriching the luminescence spectrum and the efficient use of the mechanisms of luminescence photon multiplication.

Due to a small ionic radius, Yb ions are well diluted in RE- and Y-based isomorphic structure garnets. At a high concentration, they create dimers [39–41], which bring additional capability for NIR luminescence excitation in inorganic compounds.

Compositionally disordered compounds with a garnet structure provide a great opportunity to tune disposition of electronic energy levels into bandgap of the compound and conditions for electronic excitations transfer from matrices to doping ions [42]. Therefore, we investigated a series of the quintuple garnet structure compounds in a ceramics form $(Gd, Y, Yb)_3Al_2Ga_3O_{12}$, which were solely doped with Ce or Tb. Both ions are used to create bright scintillation materials. However, their radiating states have a different location in the forbidden gap of the crystalline systems with a close composition. Samples were characterized using X-ray diffraction to demonstrate the phase homogeneity of the obtained ceramics. Furthermore, they were investigated for luminescent properties at excitation with different light sources, including synchrotrons as well as ionizing radiation. The results obtained were supported by modelling of the luminescence spectra and excitation transfer processes.

## 2. Materials and Methods

Using the procedure described in [43], a series of ceramic samples based on the two basic compositions $Gd_3Al_2Ga_3O_{12}$ (GAGG) and $Gd_{1.5}Y_{1.5}Al_2Ga_3O_{12}$ (GYAGG) were prepared. Gadolinium was replaced by Ce and Tb, and yttrium was replaced by Yb (in the absence of yttrium, ytterbium was used instead of gadolinium). Eight samples with different Yb concentrations were produced; their compositions and X-ray densities are presented in Table 1. The ceramic samples were obtained by sintering in an oxygen atmosphere at 1600 °C, which reliably excluded the localization of ytterbium ions in the divalent state. In addition, the density of the samples was determined using the method of hydrostatic weighing and was found to be >99.6% for each composition. All samples with a thickness of 1 mm were translucent.

**Table 1.** Chemical compositions, abbreviations, and calculations of the density based on the XDR data of samples $(Gd, RE)_{1.5}(Y, Yb)_{1.5}Al_2Ga_3O_{12}$ (RE = Ce, Tb).

| # | Composition | Abbreviation | Calculated Density, g/cm³ |
|---|---|---|---|
| 1a | $Gd_{2.85}Al_2Ga_{2.97}O_{12}Ce_{0.015}$ | Gd3Yb0Y0-Ce | 6.67 |
| 1b | $Gd_{2.90}Al_2Ga_{2.97}O_{12}Tb_{0.10}$ | Gd3Yb0Y0-Tb | 6.67 |
| 2 | $Gd_{1.485}Yb_{1.5}Al_2Ga_{2.91}O_{12}Ce_{0.015}$ | Gd1.5Yb1.5Y0-Ce | 6.96 |
| 3 | $Gd_{1.350}Yb_{1.5}Al_2Ga_{2.91}O_{12}Tb_{0.15}$ | Gd1.5Yb1.5Y0-Tb | 6.92 |
| 4 | $Gd_{1.485}Y_{0.75}Yb_{0.75}Al_2Ga_{2.91}O_{12}Ce_{0.015}$ | Gd1.5Yb0.75Y0.75-Ce | 6.47 |
| 5 | $Gd_{1.350}Y_{0.75}Yb_{0.75}Al_2Ga_{2.91}O_{12}Tb_{0.15}$ | Gd1.5Yb0.75Y0.75-Tb | 6.43 |
| 6 | $Gd_{1.485}Y_{1.2}Yb_{0.3}Al_2Ga_{2.91}O_{12}Ce_{0.015}$ | Gd1.5Yb0.3Y1.2-Ce | 6.18 |
| 7 | $Gd_{1.350}Y_{1.2}Yb_{0.3}Al_2Ga_{2.91}O_{12}Tb_{0.15}$ | Gd1.5Yb0.3Y1.2-Tb | 6.17 |
| 8a | $Gd_{1.485}Y_{1.5}Al_2Ga_{2.91}O_{12}Ce_{0.015}$ | Gd1.5Yb0Y1.5-Ce | 5.91 |
| 8b | $Gd_{1.400}Y_{1.5}Al_2Ga_{2.91}O_{12}Tb_{0.10}$ | Gd15Yb0Y1.5-Tb | 5.94 |

The samples were measured by X-ray powder diffraction (XRD) using a Bruker D2 PHASER with CuKα radiation in the Bragg–Brentano geometry. The intensity and position of the reflections (Figure 1) of the samples correspond to the phase of the garnet family: the cubic system of the space group Ia3d (PDF-46-0448), which indicates the incorporation of ytterbium in the garnet phase into dodecahedral positions ($RE^{3+}$) without the formation of separate secondary phases.

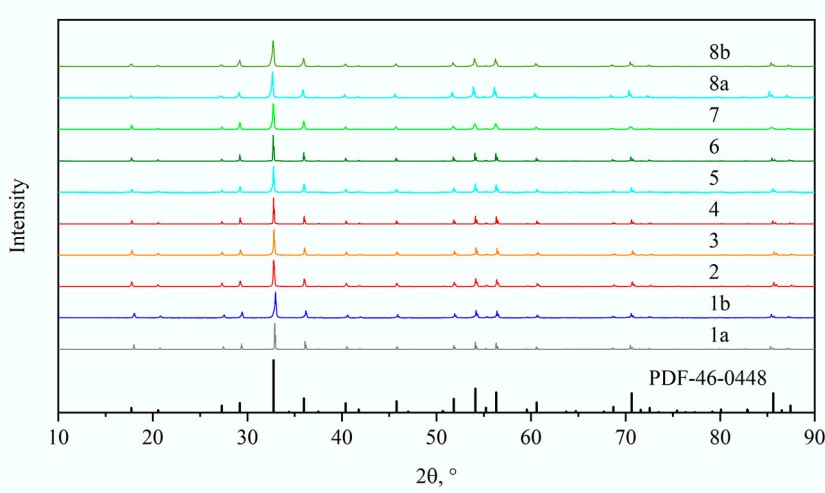

**Figure 1.** X-ray diffraction patterns of cerium/terbium-activated ceramic samples with different Gd:Yb:Y ratios.

A ceramic microstructure was studied using a Jeol JSM-7100F scanning electron microscope with a Schottky cathode and a resolution of 3 nm at an accelerating voltage of 6 kV. SEM images of representative samples were obtained in backscattered electron mode and are shown in Figure 2. All samples demonstrate garnet-type habitus.

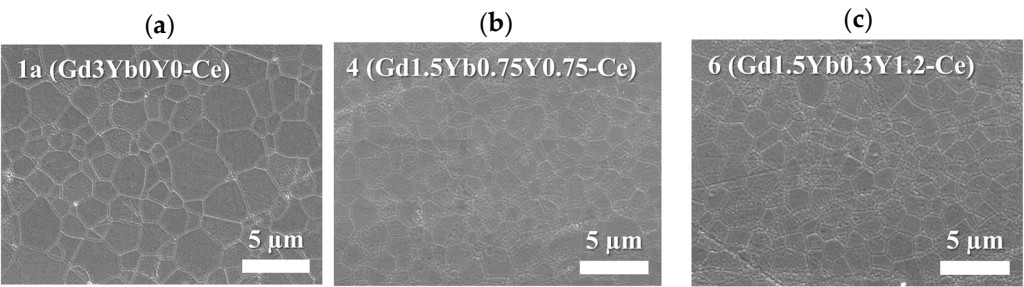

**Figure 2.** SEM images of representative (indicated) ceramic samples: (**a**)—$Gd_{2.85}Al_2Ga_{2.97}O_{12}Ce_{0.015}$, (**b**)—$Gd_{1.485}Y_{0.75}Yb_{0.75}Al_2Ga_{2.91}O_{12}Ce_{0.015}$, (**c**)—$Gd_{1.485}Y_{1.2}Yb_{0.3}Al_2Ga_{2.91}O_{12}Ce_{0.015}$.

The photoluminescence (PL) and photoluminescence excitation (PLE) spectra were measured using the Fluorolog-3 spectrofluorometer (HORIBA Scientific, USA). A liquid nitrogen-cooled InGaAs Symphony II CCD array (HORIBA Scientific, USA) was used to measure PL spectra, and R5509-73 liquid nitrogen-cooled PMT (Hamamatsu, Japan) was used to detect PLE spectra in the near-infrared (NIR) range. The measured PL and PLE spectra were corrected by the spectral sensitivity of the recording system and the distribution of the spectral density of the exciting radiation, respectively, and were expressed as the dependence of the number of quanta per unit wavelength interval ($dN/d\lambda$) on $\lambda$. PL spectra were measured in the geometry close to 45°. Samples were positioned the same way to provide comparison of the luminescence intensities.

The PL kinetics were measured using PicoQuant Fluotime 250 spectrofluorimeter under pulsed LED excitation. In addition, measurements of the temperature dependence of the luminescence spectra of the sample Gd1.5Yb0.3Y1.2-Ce (#6) were performed using synchrotron radiation at the P66 experimental station "SUPERLUMI" at PETRA III storage ring (DESY, Hamburg, Germany). The sample was installed on the cold finger of helium cryostat that allowed for the performance of measurements in the temperature region 5–300 K. Luminescence spectra were measured using Andor Kymera 328i spectrograph equipped with ANDOR Newton 920 CCD camera. For Ce-doped samples, the scintillation light yield (LY) in the visible range was evaluated by measurement of the full absorption peak position of [241]Am alpha particles as described for the measurements with translucent

or powdered samples in [44]. The measurement procedure in a current PMT mode was applied to Tb-doped samples. PMT anode current was integrated over time with a time constant of 0.1 s and its value was measured with a digital multimeter. Error of measurements was estimated to be +/−2%. The light yield of each sample was supposed to be proportional to the PMT anode current measured with the alpha source excitation after subtraction of the PMT "dark" current, measured without the alpha source.

## 3. Results and Discussion

### 3.1. Luminescence and Scintillation Properties

Figure 3 shows the samples in daylight and with illumination at two wavelengths in the UV range 365 and 250 nm. Interesting, in the samples with Ce and ytterbium the luminescence of $Ce^{3+}$ ions is completely quenched, which agrees with the results in other garnet systems [45]. The kinetics of scintillations in such samples is close to the instrumental function of PMTs used in equipment for measuring the scintillation kinetics, which has an FWHM of ~1.2 ns.

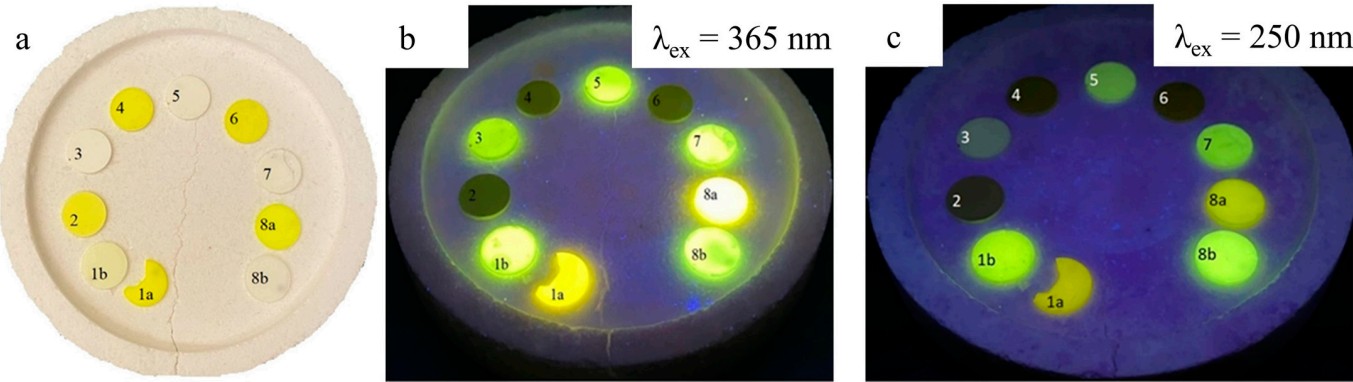

**Figure 3.** Images of the samples presented in Table 1 in daylight (**a**); excitation using UV radiation with λ = 365 nm (**b**) and λ = 250 nm (**c**).

PL spectra in the NIR range and excitation spectra of NIR luminescence of $Yb^{3+}$ ions in $Ce^{3+}$- and $Tb^{3+}$-activated samples are shown in Figure 4. Excitation spectra included bands predominantly due to intracenter transitions related to the Yb, Ce, Tb, and Gd ions and were not corrupted by defect centers and appropriate color centers as described in [46].

As seen, ytterbium ions cause an intense complex NIR PL band in both cerium- and terbium-activated samples. NIR luminescence is excited in both intra- and interconfiguration transitions of activators. In the cerium-activated samples, the PLE spectra show bands of $Ce^{3+}$ ions corresponding to transitions from the ground state to the $5d_1$ and $5d_2$ levels. A sharp rise in the excitation intensity is also observed in the spectral range shorter than 240 nm. It is caused by the long wavelength wing of the $Yb^{3+}$ CTT. Note the change in the shape of the PLE band due to the transition to the lowest excited state $5d_1$ of $Ce^{3+}$ ions in the spectral range 400–500 nm. With increasing ytterbium content, the band becomes more symmetrical. This is due to a band with a maximum near 430 nm, which is distinctively observed in PLE in Tb-activated crystals (Figure 3b). The band has a very good resonance with the interconfiguration $Ce^{3+}$ transition $4f^15d^0 \rightarrow 4f^05d^1$. The presence of resonance is a necessary condition for efficient dipole–dipole resonant transfer and electron tunneling as well [47]. To refine the mechanism of luminescence quenching of $Ce^{3+}$ ions, the temperature dependence of the luminescence spectra in the spectral range 300–1100 nm upon excitation using a synchrotron source was evaluated. The results of measurements with excitation at 225 nm (in the region of an increase in the excitation intensity) are shown in Figure 5.

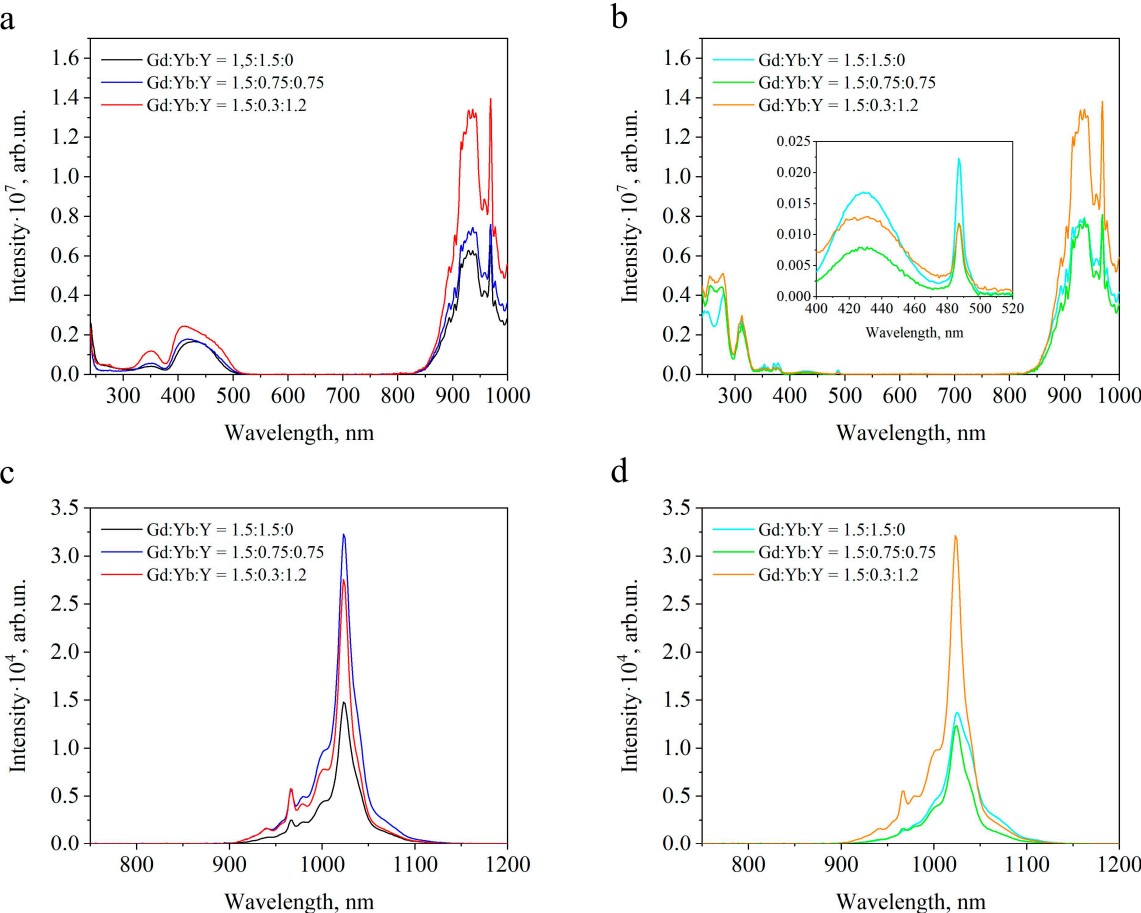

**Figure 4.** Luminescence excitation spectra of representative Ce (**a**)- and Tb (**b**)-doped samples at the registration of 1023 nm; room temperature measured luminescence spectra in the IR region at 365 nm excitation of the same Ce (**c**)- and Tb (**d**)-doped samples.

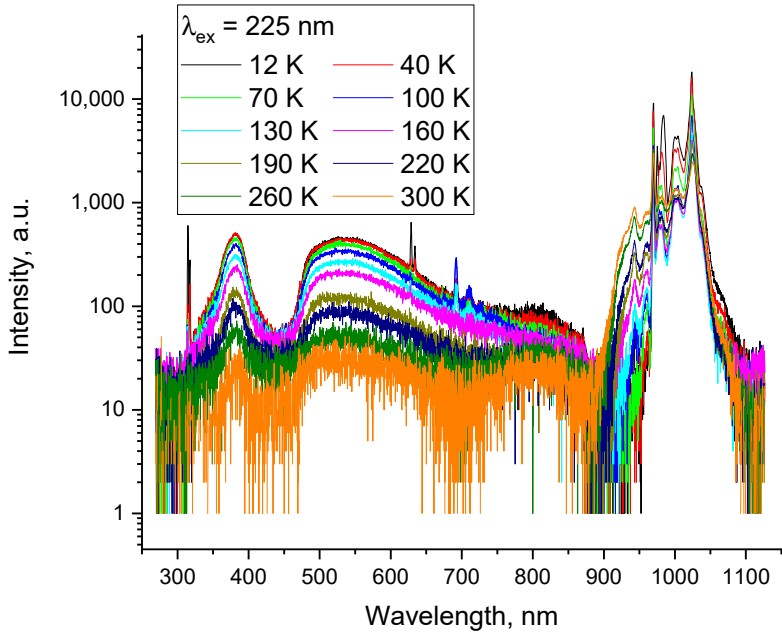

**Figure 5.** Luminescence spectra of Gd1.5Yb0.3Y1.2-Ce (#6) at excitation 225 nm at different temperatures.

The spectra show two groups of wide complex bands with maxima near 360 nm, a broad band in the spectral range 450–700 nm, and a group of bands in the NIR range due to intracenter transitions in $Yb^{3+}$ ions. The first from the short wavelength edge is $Yb^{3+}$ CTT luminescence band CTS$\rightarrow^2F_{7/2}$, whereas the band in the visible range is a superposition of the $Yb^{3+}$: CTS$\rightarrow^2F_{5/2}$, and a few $Ce^{3+}$ PL bands: $5d^14f^0\rightarrow^2F_{7/2,5/2}$, which appear in a compositionally disordered garnet-type compound. The $Gd^{3+}$ luminescence $^6P_J\rightarrow^8S_{7/2}$ is observed at ~315 nm up to 100 K. In addition, the *f-f* luminescence in the range 600–750 nm is observed due to a trace concentration of $Eu^{3+}$ ions. This luminescence is quenched above 100 K. Apparently, the $Eu^{3+}$ luminescence is quenched by Yb ions as well, and an appropriate band corresponding to the europium $^7F_6\rightarrow^5D_0$ transition is observed in NIR PLE near 490 nm (Figure 4b), indicating a sensitization.

To distinguish the temperature dependences of the intensity of PL bands of different origins, the spectrum was approximated by a set of single line shape functions $G(w)$ presented in [48]. A satisfactory result of deconvolution is achieved by decomposing the luminescence band into at least two doublets of bands due to $Ce^{3+}$ ions with a splitting value of 0.25 eV and three overlapping CTS$\rightarrow^2F_{5/2}$ bands. The appearance of three CTS$\rightarrow2F_{5/2}$ bands of $Yb^{3+}$ ions occupying dodecahedral positions is caused by three main combinations of localization of Ga and Al ions in the surrounding octahedral and tetrahedral coordination in the compositionally disordered GYAGG compound, as demonstrated in [48]. The CTT luminescence band CTS$\rightarrow^2F_{7/2}$ also looks nonelementary. Unsmooth variation in the intensity is observed in the region of the long-wave and the short-wave wings. It seems that it consists of three bands as well.

The spectral shape of PL in the range 450–700 nm can be described as

$$F(\omega) = A_{Ce}F_{Ce}(\omega) + A_1G(\omega - \omega_1) + A_2G(\omega - \omega_2) + A_3G(\omega - \omega_3) \tag{1}$$

where $G(\omega) = \int_{-\infty}^{+\infty} e^{i\omega t - S((1-cos(\Omega t))cth(\frac{\hbar\Omega}{2k_BT})-isin(\Omega t))} dt$, S—Huang–Rice factor (taken to be 6, as described in [49]), $k_B$—Boltzmann constant, T—temperature, $\Omega = 53$ meV—LO phonon frequency, $F_{Ce}(\omega)$—band shape provided by $Ce^{3+}$ ions [43], $A_i$ are weight multipliers. Note that the line shape function $G(\omega)$ satisfies the following normalization condition: $\int_0^\infty G(\omega)dw = 1$. Here, $\omega$ is the light circular frequency. The line $Ce^{3+}$ is normalized in a similar way: $\int_0^\infty F_{Ce}(\omega)d\omega = 1$. An example of deconvolution of the band in the spectral range of 450–700 nm at a temperature of 160 K is shown in Figure 6.

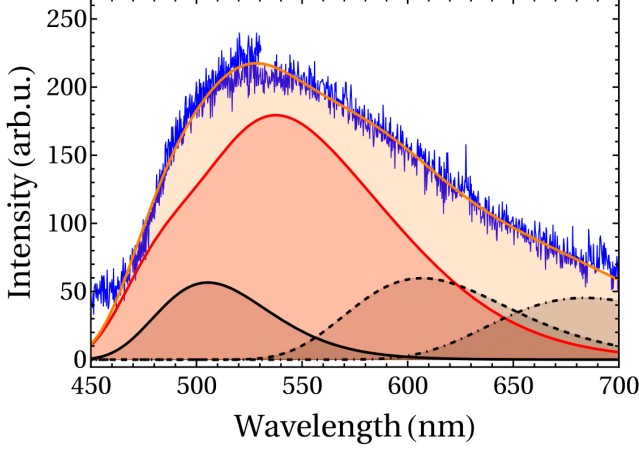

**Figure 6.** Deconvolution of the experimental luminescence band contour (blue line) in the spectral region 450–700 nm into the total band of $Ce^{3+}$ ions (red line) and three CTS$\rightarrow^2F_{5/2}$ bands (black lines) at a temperature of 160 K. The total contour of the band due to transitions with an involvement of $Ce^{3+}$ and $Yb^{3+}$ ions is superimposed on the experimental curve (orange line).

Using the results of deconvolution of the spectra measured at different temperatures, the temperature dependence of the integral intensities of the $Ce^{3+}$- and $Yb^{3+}$-related bands was defined (Figure 7a). Figure 7b shows the temperature dependence of the $Yb^{3+}\ ^2F_{5/2}\rightarrow^2F_{7/2}$ PL-integrated intensity. It shows a growing intensity at a temperature ~100 °C, which corresponds to the start of $Ce^{3+}$ PL temperature quenching. It indicates a sensitization of the $Yb^{3+}$ luminescence.

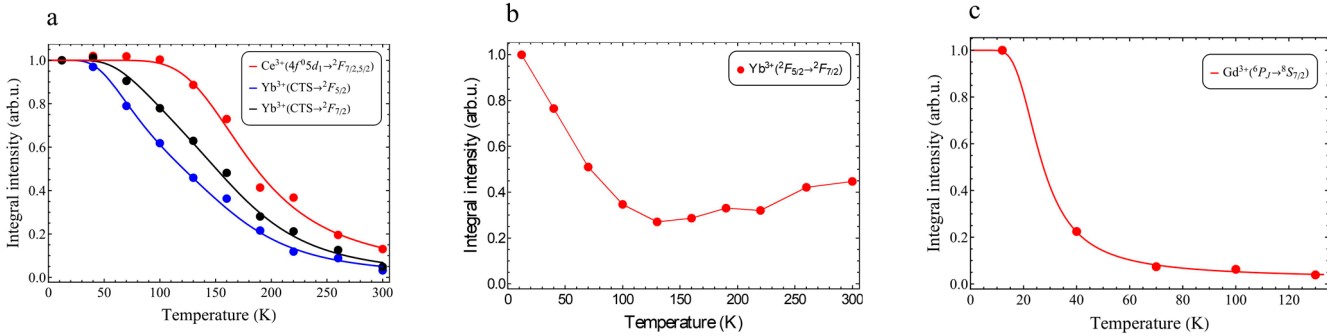

**Figure 7.** Temperature dependence of the integrated luminescence intensity of the bands in the Gd1.5Yb0.3Y1.2-Ce (#6) sample: (**a**)—$Ce^{3+}$: $4f^05d_1\rightarrow^2F_{7/2,5/2}$ (red), $Yb^{3+}$: CTS$\rightarrow^2F_{5/2}$ (blue), and CTS$\rightarrow^2F_{7/2}$ (black); (**b**)—intracenter $Yb^{3+}$ NIR luminescence; (**c**)—$Gd^{3+}$: $^6P_J\rightarrow^8S_{7/2}$. Solid lines at the panels (**a**,**c**) are the approximating curves.

An approximation of the temperature dependences is shown in Figure 7. Parameters of the approximation were obtained within the model for the $Yb^{3+}$ CTT luminescence bands [12]:

$$I(T) = I_0 \frac{W_{rad}}{W_{rad} + c_1 e^{-\frac{E_1}{k_B T}} + c_2 e^{-\frac{E_2}{k_B T}}} \qquad (2)$$

where $W_{rad}$ is the probability of a radiative transition from CTS to $^2F_{7/2}$, $c_1 e^{-\frac{E_1}{k_B T}}$ is the probability of a thermally activated transition from CTS to the $^2F_{5/2}$ level (nonradiative transition), $c_2 e^{-\frac{E_2}{k_B T}}$—CTS ionization probability, $c_i$—frequency factor, $E_i$—activation energy.

For the luminescence of $Ce^{3+}$ ions, we used the single-exponential approximation:

$$I(T) = I_0 \frac{W_{rad}}{W_{rad} + c e^{-\frac{E}{k_B T}}} \qquad (3)$$

where $c e^{-\frac{E}{k_B T}}$—probability of a nonradiative transition, c is the frequency factor, E is the activation energy. The coefficients obtained as a result of the approximation of the temperature dependences of the integrated intensity are presented in Table 2. Activation energy $E_1$ of the luminescence transitions are quite similar indicating close mechanisms of their quenching.

**Table 2.** Approximation coefficients for the temperature dependences of the integrated intensity of the luminescence bands defined in Gd1.5Yb0.3Y1.2-Ce (#6) sample under synchrotron excitation.

| | $\frac{c_1}{W_{rad}}$ | $E_1$ | $\frac{c_2}{W_{rad}}$ | $E_2$ |
|---|---|---|---|---|
| $Ce^{3+}$:$4f^05d_1\rightarrow^2F_{7/2,5/2}$ | 133.0 | 0.08 eV | | |
| $Yb^{3+}$: CTS$\rightarrow^2F_{5/2}$ | 545.0 | 0.1 eV | 5.45 | 0.026 eV |
| $Yb^{3+}$: CTS$\rightarrow^2F_{7/2}$ | 557.0 | 0.092 eV | 5.57 | 0.019 eV |

The $Gd^{3+}\ ^6P_J\rightarrow^8S_{7/2}$ luminescence temperature dependence curve (Figure 6c) was approximated similarly to the case of $Ce^{3+}$ luminescence. In fact, it shows a behavior that is close to the results described in [50]. However, the activation energy was defined to

be 9.5 meV, which is two times less than in GAGG:Ce [51]. Most likely, this difference is caused by the additional quenching effect by Yb ions in the matrix.

A complete quenching of the luminescence of $Tb^{3+}$ ions does not occur. The $Yb^{3+}$ luminescence excitation spectra show both interconfigurational transitions of $Tb^{3+}$ ions near 318 and 275 nm and $f \rightarrow f$ transitions in the spectral range of 340–390 nm [32]. Note that the $^7F_0 \rightarrow {}^5D_4$ transition of $Tb^{3+}$ ions is located in between the CTT bands of $Yb^{3+}$ ions, i.e., there is no resonance. Therefore, with an increase in the concentration of ytterbium in the compound, a slight decrease in the decay time of photoluminescence occurs (Table 3). However, such a slight reduction in the luminescence decay constant is in poor agreement with the course of the decrease in the integral intensity of the luminescence of $Tb^{3+}$ ions, which decreases by an order of magnitude with an increase in the ytterbium concentration in the range X = 0.3 to 1.5. The decrease in the intensity of $Tb^{3+}$ scintillations is even two times greater. Apparently, the $Yb^{3+}$ ions, due to the efficient capture of electrons, contribute to the nonradiative relaxation of Frenkel excitons localized in the gadolinium sublattice of the crystal. Such excitons, as shown in [32], make a significant contribution to the high scintillation yield in GYAGG:Tb.

**Table 3.** Photoluminescence (546 nm) decay constants for terbium-activated samples at 315 nm excitation and relative changes in the integrated intensity of $Tb^{3+}$ luminescence and scintillations depending on the Yb content in the sample.

| Sample | Photoluminescence Decay Constant at Excitation 315 nm $\tau$, ms | Relative Yield of $Tb^{3+}$ Photoluminescence 380–750 nm at Excitation 275 nm, rel. un | Relative LY of Scintillation, Rel. un |
|---|---|---|---|
| Gd1.5Yb1.5Y0-Tb | 2.27 | 0.1 | 0.05 |
| Gd1.5Yb0.75Y0.75-Tb | 2.67 | 0.4 | 0.07 |
| Gd1.5Yb0.3Y1.2-Tb | 2.82 | 0.9 | 0.25 |
| Gd15Yb0Y1.5-Tb | 2.91 | 1.0 | 1.0 |

### 3.2. $Ce^{3+}$ and $Tb^{3+}$ Luminescence Quenching Models

For heterovalent ytterbium ion in crystals, one should distinguish between ions that are localized in the lattice at the synthesis of the material and at the capture of the nonequilibrium carrier.

In the garnet structure, $Yb^{3+}$ ions, isomorphically substitute yttrium or RE ions. Its charge state is stable for an arbitrarily long time in the absence of a specific external effect on such a crystal. The question of the location of the levels of the states $^2F_{5/2}$ and $^2F_{7/2}$ relative to the band gap remains rather unexplored. The authors of [52] placed both levels in the valence band. This disposition contradicts the fact that $Yb^{3+}$ ions have the highest electron affinity among trivalent RE ions. Following the zigzag diagrams [14], and accounting for the appearance of the $Gd^{3+}$ ground state ~1 eV below the top of the valence band in a $Gd_3Al_2Ga_3O_{12}$ crystal of close composition [38], we concluded that the depth of the ground state $^2F_{7/2}$ of $Yb^{3+}$ ions is close to a half of the energy gap between the $^2F_{7/2}$ and $_2F_{5/2}$ states, i.e., the $^2F_{5/2}$ excited state is 0.6 eV above the top of the valence band.

A capture of an electron by $Yb^{3+}$ ions in the garnet structure leads to the formation of $Yb^{2+}$ ions, the charge state of which is metastable. Stabilization of ions with a metastable charge is accompanied by local distortions of the matrix to compensate for the excessive negative charge. Such a center has a depth of about 1 eV relative to the bottom of the conduction band in an Al/Ga garnet crystal [53] and causes, upon its thermal ionization into the conduction band, persistent luminescence of $Ce^{3+}$ ions. There is a debated point of view in the literature about the formation of stable $Ce^{4+}$-$Yb^{2+}$ pairs at the doping of ytterbium-containing compounds with Ce ions. It is obvious that such pairs can arise under certain technological conditions, in particular, at co-doping with heterovalent impurities. For example, the formation of pairs with $Yb^{2+}$ ions was demonstrated in vacuum-grown $Y_3Al_5O_{12}$ co-doped with heterovalent Yb and Fe ions [15,29].

Finally, one should also take into account the dynamic charge state of ions, which occurs for a limited period of time. For example, an electron of the $Ce^{3+}$ ion, when excited to $5d_{2-5}$ levels, which are localized above the bottom of the conduction band, can be delocalized into the conduction band with the formation of the dynamic state $Ce^{4+} + e^-$. The role of such dynamic states in the scintillation process is considered in [54]. Obviously, such dynamic states can also arise for other RE upon excitation to high-lying levels or CCT. Another example is an excitation of the $O^2 \rightarrow Yb^{3+}$ CTT, when a dynamic charge state of $Yb^{2+}$ ion arises and rapidly relaxes. Looking for the mechanisms of the quenching, we obviously deal with the dynamic states of the $Yb^{2+}$ in the crystals having charge-stable $Yb^{3+}$ ions in the matrix.

In the case of $Ce^{3+}$ ions, an electron tunneling from $Ce^{3+}$ to $Yb^{3+}$ drives the quenching of the cerium luminescence by $Yb^{3+}$ ions. Figure 8a,b provide configuration curves related to sites involved in this process. In addition to cerium *f*-and lowest *d*-states usual for the energy diagram of this ion, a curve for the $Ce^{4+} +e$ state is presented corresponding to cerium ionization with the electron transfer to the conduction band.

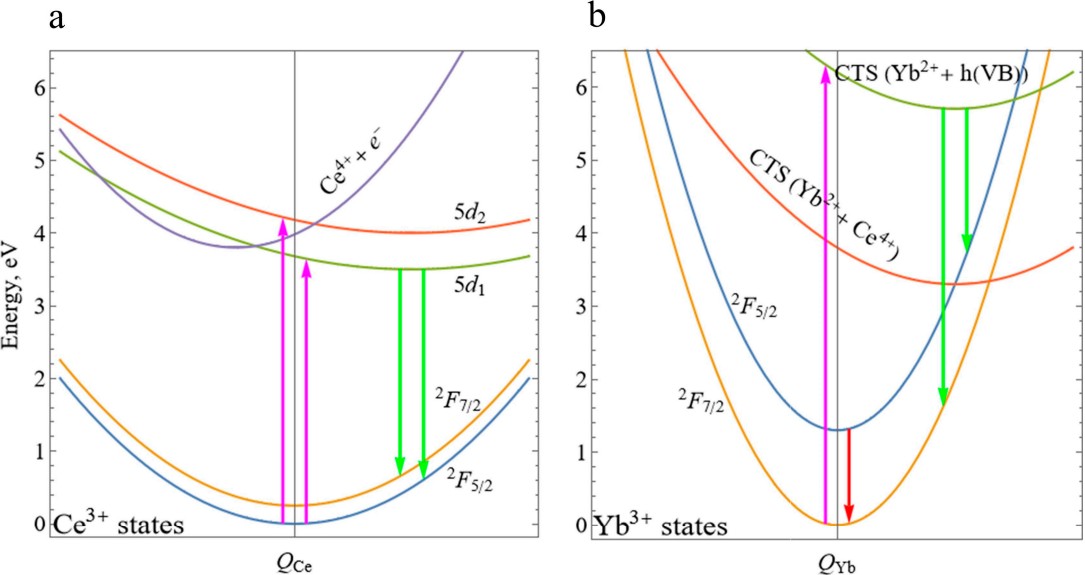

**Figure 8.** Configuration diagrams for isolated $Ce^{3+}$ (**a**), $Yb^{3+}$ (**b**) ions. Electronic transitions caused by photon absorption and emission are shown by vertical arrows.

For the concentrations of Yb ions in the samples under consideration, the probability to find at least one ytterbium ion in the nearest neighborhood of any cerium ion at a distance of 3.75 Å is high. Such a distance favors electron tunneling process from cerium to ytterbium. Ytterbium ion in the close vicinity of excited cerium ion can affect relaxation of the latter in the following way: $Ce^{3+*} + Yb^{3+}$ (stage 1)$\rightarrow Ce^{4+} + Yb^{2+}$ (stage 2)$\rightarrow Ce^{3+} + Yb^{3+}$(stage 3). The total excitation energy in stage 1 is ~3 eV higher than that in stage 3 due to the rearrangement of the respective lattice sites using phonon emission.

The configuration potentials of the complex consisted of the closely spaced Ce-Yb ions is shown in Figure 9. The total energy of this complex center is plotted as a function of two configuration coordinates: one for the ions in the vicinity of $Ce^{3+}$ ($Q_{Ce}$) and the other one of $Yb^{3+}$ ($Q_{Yb}$). Five energy surfaces in the figure correspond to the states described in the legend on the right of the panel. Point A corresponds to the ground state of the system, i.e., to the equilibrium environment of both ions. Point B is reached when the cerium ion is excited to $5d_1$ state followed by the lattice's relaxation. Point C stands for the relaxed neighborhood of CTS ($Ce^{4+}$-$Yb^{2+}$) following electron tunneling from cerium to ytterbium (not to be confused with CTS of isolated $Yb^{3+}$).

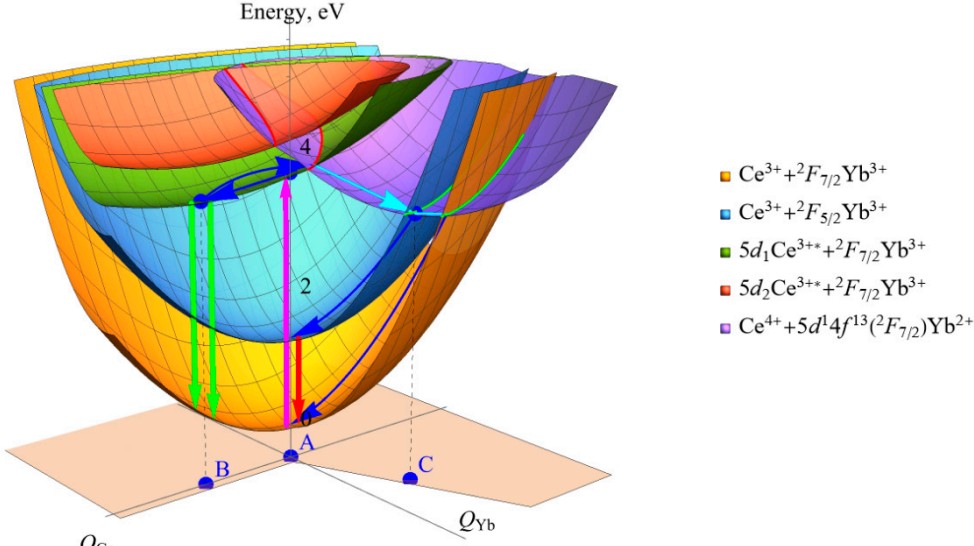

**Figure 9.** Configuration diagram for Ce + Yb complex. Electronic transitions caused by photon absorption and emission are shown by vertical arrows. Points A, B, and C correspond to the minima of the ground state, relaxed $Ce^{3+*}$, and relaxed position of $Ce^{4+} + Yb^{2+}$ of the dynamic complex, respectively. Red and green lines indicate intersection of configuration surfaces where electron passes from cerium to ytterbium (red) and returns to cerium (green). Blue arrows indicate relaxation of the system in $Ce^{3+} + Yb^{3+}$ state, whereas cyan arrows show the relaxation pathway in dynamic $Ce^{4+} + Yb^{2+}$ state.

Intersection (crossing) of energy surfaces of two different charge states of the ion pairs ($Ce^{3+*}+Yb^{3+}$ and $Ce^{4+}+Yb^{2+}$) is indicated by red curves and corresponds to the recharging of the RE ions. The transition between the surfaces $Ce^{3+}+Yb^{3+}$ and $Ce^{4+}+Yb^{2+}$ (cerium ion is no longer excited) is shown by green curves. Blue and cyan arrows mark the non-radiative relaxation channels existing in the complex center. One of the possible relaxation states of $Yb^{3+}$ results in the emission of the NIR *f-f* emission. A relatively narrow PLE of NIR luminescence with a maximum of 430 nm (2.9 eV) and a half-width of 0.2 eV is detected, whose intensity depends on ytterbium content (see thy inset in Figure 4b). In the studied range of Yb concentrations, the Yb ion has a high probability of getting another Yb ion in the first cationic coordination sphere. Most likely, this band is due to the transition in $Yb^{3+}$-$Yb^{3+}$ pairs [39,40], although a rigorous proof requires measurement of the absorption spectra, which is quite difficult to achieve in translucent samples. Dimers provide an insignificant contribution to the intensity of excitation of the NIR luminescence in comparison with Ce ions or intracenter excitation of $Yb^{3+}$ ions. Nevertheless, the fact that this transfer mechanism appears in samples with a high ytterbium content is noteworthy. Note the good resonance of the 430 nm band and the lower interconvolutional transition of the $Ce^{3+}$ ions. When dimers are excited, they can transfer the excitations to Ce ions; therefore, they can contribute to the NIR luminescence excitation through this channel as well.

When scintillation is excited, the gadolinium subsystem in the matrix collects Frenkel-type excitons [37]. Since the concentration of Ce is much lower than that of ytterbium in the studied samples, $Yb^{3+}$ ions effectively compete with $Ce^{3+}$ ions in capturing electronic excitations. The energy of the Frenkel exciton is likely to be exchanged for two NIR quanta with the participation of phonons during recombination.

In the samples activated with $Tb^{3+}$ ions, the quenching processes differ. Note, the $Yb^{3+}$ NIR PLE shows both interconfigurational and $f \rightarrow f$ intraconfigurational transitions of $Tb^{3+}$ ion. However, the transfer from the $^5D_4$ state, as seen from a minor change of the luminescence decay component (Table 3), is unlikely. On the contrary, the upper radiation state $^5D_3$, having decay constant ~150 µs in ytterbium-free samples [55], is effectively devastated by transfer to Yb ions. This process diminishes nonradiative relaxation $^5D_3 \rightarrow ^5D_5$ and signifi-

cantly decreases the yield of the luminescence when excited in upper *df* mixed states. When scintillations are excited, the perfect overlapping of $^8S{\rightarrow}^6P$ ($Gd^{3+}$) and $^7F_0{\rightarrow}4f^7[^8S]5d_1$ (high spin) ($Tb^{3+}$) levels is activated. At comparable concentrations, both the terbium and ytterbium subsystems will be in roughly equal conditions to catch Frenkel-type exciton from the Gd-subsystem. Nevertheless, some fraction of electronic excitations localized at $Tb^{3+}$ ions might be converted in the $Yb^{3+}$ luminescence by the quantum cutting for two NIR photons. Thus, doping with Tb ions looks promising for arranging a wide-spectrum scintillation, from UV to NIR, in $(Gd, Y, Yb)_3Al_2Ga_3O_{12}$ crystalline material.

### 4. Conclusions

The processes of luminescence and scintillation quenching of Ce and Tb activators in $(Gd, Y, Yb)_3Al_2Ga_3O_{12}$ compounds containing ytterbium in the matrix with X = 0.3–1.5 have been studied. It was shown that the photoluminescence quenching of cerium ions is provided both by interactions with closely spaced ytterbium $Yb^{3+}$ ions and, probably, with pairs of these ions. However, the dominant quenching mechanism is the tunneling of electronic excitations from excited $Ce^{3+*}$ ions to ytterbium ions with the formation of the $Ce^{4+}$-$Yb^{2+}$ dynamic state. The combination of these processes leads to complete quenching of photoluminescence and scintillations at room temperature. In the case of activation by terbium ions, the luminescence is not completely quenched; however, sensitization of the NIR luminescence of $Yb^{3+}$ ions upon excitation of $Tb^{3+}$ ions into both intra- and interconfiguration transitions has been established. The high efficiency of excitation of the luminescence of terbium ions in disordered systems with a garnet structure and the possibility of the quantum cutting effect in the crystalline system make it possible to create highly efficient sources of optical radiation in a wide range, including NIR radiation when excited using ionizing radiation.

**Author Contributions:** Conceptualization, M.K. and V.R.; methodology, V.D.; validation, I.K. (Irina Kamenskikh) and D.K.; investigation, I.K. (Ilia Komendo), A.R. and G.M.; writing—original draft preparation, M.K. and Y.T.; writing—review and editing, V.R. and A.V. All authors have read and agreed to the published version of the manuscript.

**Funding:** This research received no external funding.

**Institutional Review Board Statement:** No Institutional Review Board Statement is required.

**Informed Consent Statement:** Not applicable.

**Data Availability Statement:** No new and additional data are available.

**Acknowledgments:** The authors at NRC "Kurchatov Institute" and Moscow State University acknowledge support from the Russian Ministry of Science and Education, agreement no. 075-15-2021-1353 dated 12 October 2021. Analytical research was carried out using equipment from the Research Chemical and Analytical Center NRC "Kurchatov Institute" Shared Research Facilities under the project's financial support by the Russian Federation, represented by The Ministry of Science and Higher Education of the Russian Federation, agreement no. 075-15-2023-370 dd. 22 February 2023.

**Conflicts of Interest:** The authors declare no conflict of interest.

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
