# Peer review of "On the Quenching Mechanism of Ce, Tb Luminescence and Scintillation in Compositionally Disordered (Gd, Y, Yb)3Al2Ga3O12 Garnet Ceramics"

_photonics, doi:10.3390/photonics10060615_

Round 1

Reviewer 1 Report

This manuscript reports luminescence quench of Ce, Tb in (Gd,Y,Yb)3Al2Ga3O12. Authors carefully compared photoluminescence spectra, like sample element ratio, temperature. This work is well written and organized overall but also lack illustration in some parts that may lead to confusion. This paper may be accepted after major revision as follow,

1.     Please provide more illustration to Figure3 a, b. Based on presented information, I think authors claim a, b images are PLE spectra. It looks strange to me as intensity increase when excited by 900 nm.

2.     Figure3. Authors showed three sample photoluminescence spectra. However, spectrum of Yb:Y ratio 1.5:0 sample is close to Yb:Y ratio 0.75:0.75. So, basically, authors use only two photoluminescence spectra to illustrate that the quenching is caused by the increase of Yb ratio, I think more samples of different Yb:Y ratio can make the conclusion more solid.

3.     Can authors provide SEM, TEM or AFM images of samples?

4.     Figure 6 should be better drawn as mark a is even in another page.     

5.     Figure 6 b, can authors explain why the intensity decrease and then increase?

Author Response

The Reviewer comment

Authors response

This manuscript reports luminescence quench of Ce, Tb in (Gd,Y,Yb)3Al2Ga3O12. Authors carefully compared photoluminescence spectra, like sample element ratio, temperature. This work is well written and organized overall but also lack illustration in some parts that may lead to confusion. This paper may be accepted after major revision as follow,

1.   1. Please provide more illustration to Figure3 a, b. Based on presented information, I think authors claim a, b images are PLE spectra. It looks strange to me as intensity increase when excited by 900 nm. 

Thank you very much for reviewing the manuscript.

The figure caption in Fig. 4 (former  Fig.3) is rewritten. To avoid overloading the figures, we included the spectra of the representative samples.

2. Figure3. Authors showed three sample photoluminescence spectra. However, spectrum of Yb:Y ratio 1.5:0 sample is close to Yb:Y ratio 0.75:0.75. So, basically, authors use only two photoluminescence spectra to illustrate that the quenching is caused by the increase of Yb ratio, I think more samples of different Yb:Y ratio can make the conclusion more solid. 

We agree with the Reviewer that more data could provide more solid conclusions. In this article, we investigated the influence of Yb on the sample’s properties in the range of concentrations x = 1.5-0.3. The tata obtained provided reasonable information to generalize results and discuss trends. For the moment, we are working to  prepare samples with a smaller Yb content in the matrix host, particularly in the Ce-doped samples. These results will be the subject of a separate publication.

3.Can authors provide SEM, TEM or AFM images of samples?

 SEM images of representative samples were included. Now it is Fig.2.

4. Figure 6 should be better drawn as mark a is even in another page.  

Figure 7 (former figure  6) is remade accordingly.

5. Figure 6 b, can authors explain why the intensity decrease and then increase?

We modified an appropriate  sentence in the text to explain such behavior.

It shows a growing intensity at a temperature ~ 100 °C, which corresponds to the start of Ce3+ PL temperature quenching. It indicates a sensitization of the Yb3+ luminescence.

Reviewer 2 Report

Referee report on “On the quenching mechanism of Ce, Tb luminescence and scintillation in compositionally disordered (Gd, Y, Yb)3Al2Ga3O12  garnet ceramics”

This is a rather interesting and good paper that certainly can be recommended for publication, but clarifying and detailing some parts of the text.

Fig. 3. For a correct assessment of these materials, it would be useful to present the corresponding luminescence spectra under X-ray or electron irradiation.

Furthermore, how important and critical is radiation damage for a given type of compounds.

Comparing the excitation spectra, it was useful to evaluate the effects of possible radiation damage, since radiation defects are known to absorb in the region of 200-350 nm, as it is known for similar compounds:

Pankratova, V., Skuratov, V. A., Buzanov, O. A., et al. (2022). Radiation effects in Gd3(Al, Ga)5:O12: Ce3+ single crystals induced by swift heavy ions. Optical Materials: X16, 100217. https://doi.org/10.1016/j.omx.2022.100217

Lushchik, A., Lushchik, C., Popov, A. I., et al (2016). Influence of complex impurity centres on radiation damage in wide-gap metal oxides. Nuclear Instruments and Methods in Physics Research Section B: Beam Interactions with Materials and Atoms374, 90-96. https://doi.org/10.1016/j.nimb.2015.07.004

Lushchik, A., Kärner, T., Lushchik, C et al  (2012). Electronic excitations and defect creation in wide-gap MgO and Lu3Al5O12 crystals irradiated with swift heavy ions. Nuclear Instruments and Methods in Physics Research Section B: Beam Interactions with Materials and Atoms286, 200-208. https://doi.org/10.1016/j.nimb.2011.11.016

Author Response

The Reviewer comment

Authors response

1 This is a rather interesting and good paper that certainly can be recommended for publication, but clarifying and detailing some parts of the text.

Fig. 3. For a correct assessment of these materials, it would be useful to present the corresponding luminescence spectra under X-ray or electron irradiation.

Furthermore, how important and critical is radiation damage for a given type of compounds.

Thank you very much for reviewing the manuscript.

We agree with the Reviewer that XRL results might top off the article. However, our article is focused on the mechanism of the electronic excitation transfer in a complex garnet system containing Yb ions. Nevertheless, we find the Reviewer idea quite interesting for future research.

Comparing the excitation spectra, it was useful to evaluate the effects of possible radiation damage, since radiation defects are known to absorb in the region of 200-350 nm, as it is known for similar compounds:

Pankratova, V., Skuratov, V. A., Buzanov, O. A., et al. (2022). Radiation effects in Gd3(Al, Ga)5:O12: Ce3+ single crystals induced by swift heavy ions. Optical Materials: X16, 100217.https://doi.org/10.1016/j.omx.2022.100217

Lushchik, A., Lushchik, C., Popov, A. I., et al (2016). Influence of complex impurity centres on radiation damage in wide-gap metal oxides. Nuclear Instruments and Methods in Physics Research Section B: Beam Interactions with Materials and Atoms374, 90-96.https://doi.org/10.1016/j.nimb.2015.07.004

Lushchik, A., Kärner, T., Lushchik, C et al  (2012). Electronic excitations and defect creation in wide-gap MgO and Lu3Al5O12 crystals irradiated with swift heavy ions. Nuclear Instruments and Methods in Physics Research Section B: Beam Interactions with Materials and Atoms286, 200-208. https://doi.org/10.1016/j.nimb.2011.11.016

The authors thank the Reviewer for interesting references. The list of references is updated, and appropriate sentences are included in the article.

Reviewer 3 Report

No comments

minor revision

Author Response

Reviewer 3

The Reviewer comment

Authors response

Comments and Suggestions for Authors.

No comments.

Thank you very much for reviewing the manuscript.

Comments on the Quality of English Language

Minor revision

Checked  by the language carrier.

Round 2

Reviewer 1 Report

authors' responde that "SEM images of representative samples were included. Now it is Fig.2.", It means SEM is already included in version 1 manucript? There is no SEM images in version 1 manuscript.